# Interferon-γ Preferentially Promotes Necroptosis of Lung Epithelial Cells by Upregulating MLKL

**DOI:** 10.3390/cells11030563

**Published:** 2022-02-06

**Authors:** Qin Hao, Sreerama Shetty, Torry A. Tucker, Steven Idell, Hua Tang

**Affiliations:** Department of Cellular and Molecular Biology, The University of Texas Health Science Center at Tyler, Tyler, TX 75708, USA; qin.hao@uthct.edu (Q.H.); sreerama.shetty@uthct.edu (S.S.); torry.tucker@uthct.edu (T.A.T.); steven.idell@uthct.edu (S.I.)

**Keywords:** IFNγ, necroptosis, acute lung injury, lung epithelial cells, MLKL

## Abstract

Necroptosis, a form of programmed lytic cell death, has emerged as a driving factor in the pathogenesis of acute lung injury (ALI). As ALI is often associated with a cytokine storm, we determined whether pro-inflammatory cytokines modulate the susceptibility of lung cells to necroptosis and which mediators dominate to control necroptosis. In this study, we pretreated/primed mouse primary lung epithelial and endothelial cells with various inflammatory mediators and assessed cell type-dependent responses to different necroptosis inducers and their underlying mechanisms. We found that interferon-γ (IFNγ) as low as 1 ng/mL preferentially promoted necroptosis and accelerated the release of damage-associated molecular patterns from primary alveolar and airway epithelial cells but not lung microvascular endothelial cells. Type-I IFNα was about fifty-fold less effective than IFNγ. Conversely, TNFα or agonists of Toll-like receptor-3 (TLR3), TLR4, TLR7 and TLR9 had a minor effect. The enhanced necroptosis in IFNγ-activated lung epithelial cells was dependent on IFNγ signaling and receptor-interacting protein kinase-3. We further showed that necroptosis effector mixed lineage kinase domain-like protein (MLKL) was predominantly induced by IFNγ, contributing to the enhanced necroptosis in lung epithelial cells. Collectively, our findings indicate that IFNγ is a potent enhancer of lung epithelial cell susceptibility to necroptosis.

## 1. Introduction

Acute respiratory distress syndrome (ARDS) is a form of life-threatening respiratory failure that is associated with high morbidity and mortality [1,2]. ARDS can develop following various types of acute lung injuries (ALI) resulting from pulmonary (e.g., pneumonia, aspiration) or nonpulmonary (e.g., sepsis, pancreatitis, trauma) insults [2]. Globally, ARDS affects approximately 3 million patients annually and accounts for 10% of intensive care unit admissions [3]. Due to the ongoing coronavirus disease 2019 (COVID-19) pandemic, the number of patients diagnosed with ARDS/ALI increased substantially [4,5]. Clinically, ALI/ARDS manifests as hypoxemia, lung edema, decreased lung compliance and diffuse pulmonary infiltrates [1,2,4]. Despite decades of efforts on research, the pathogenesis of ALI/ARDS is not fully elucidated and treatment options are limited as a result. Supportive care with mechanical ventilation remains the mainstay of management for ALI/ARDS [2,4]. 

The pathological findings of ALI in the acute phase include the disruption of the alveolar epithelial and endothelial barrier, recruitment of leukocytes and protein-rich fluid in the alveolar space [1]. Inflammation is an important component to the pathogenesis of ALI/ARDS. Previous studies indicate that inhibition of the chemokine CXC–CXCR receptor axis prevents neutrophil recruitment and tissue injury in several sterile inflammation-induced ALI animal models [1]. A recent study shows that palmitoylethanolamide inhibits ALI in mice induced by endotoxin lipopolysaccharide (LPS) and may represent a potential adjuvant anti-inflammatory therapeutic for treating lung injury [6]. Besides the involvement of inflammation, necroptosis, a form of programmed lytic cell death, has emerged as a driving factor in the pathogenesis of ALI [7,8,9,10]. Necroptosis causes cell membrane rupture, and triggers and amplifies inflammation via the release of damage-associated molecular patterns (DAMPs), through which cell death and inflammation promote exaggerated tissue injury and organ dysfunction [11,12]. Necroptosis can be induced by the TNF superfamily, Toll-like receptor 3 (TLR3) or TLR4 ligands when caspases are inhibited and/or when the ubiquitin ligases cIAP1/2 (cellular inhibitor of apoptosis protein) are degraded [13,14]. Necroptosis is executed by receptor-interacting protein kinase-3 (RIPK3) and the downstream effector mixed lineage kinase domain-like protein (MLKL). It has been shown that serum levels of RIPK3 are elevated in critically ill patients and are associated with the development of ALI/ARDs in sepsis, trauma and COVID-19 patients [15,16,17]. Recent experimental studies have demonstrated that inhibition of necroptosis confers protection against ALI induced by LPS [18,19,20], mechanical ventilation [21], sepsis/systemic inflammatory response syndrome [22,23,24], respiratory syncytial virus infection [25], bacterial pneumonia [26,27,28], trauma [29], and blood transfusion [30]. Hence, the targeting of necroptosis or its modulators holds significant promise for the treatment of ALI/ARDS.

The intracellular signaling pathways associated with necroptosis have been well studied [13,14]. However, the extracellular modulators of necroptosis including enhancers and suppressors are not known. As ALI is often associated with a cytokine storm, it remains unclear as to whether the pro-inflammatory cytokines control the susceptibility of lung cells to necroptosis. The critical cytokines that modulate the necroptosis of different types of lung parenchymally derived cells are likewise poorly understood. To address these gaps in current knowledge, we pretreated/primed mouse primary lung epithelial and endothelial cells with various inflammatory mediators and investigated the cell type-dependent responses to necroptosis and the underlying mechanisms. We provide compelling evidence that IFNγ preferentially among various inflammatory mediators promotes necroptosis in primary alveolar and airway epithelial cells but not in lung microvascular endothelial cells. This process involves the upregulation of necroptosis effector MLKL in lung epithelial cells. Our findings provide a mechanistic link and a novel interpretation of the contribution of IFNγ to the pathogenesis of ALI.

## 2. Materials and Methods

### 2.1. Antibodies and Reagents

GSK872 (no. 530389) and JAK kinase inhibitor-1 (no. 420099) were from EMD Millipore (Burlington, MA, USA). GSK843 (HY-125402) and GW806742X (no. HY-112292) were from MedChemExpress (Monmouth Junction, NJ, USA). Vinculin (no. V9131) antibody and LPS (*Escherichia coli* 0111:B4) were from Sigma (St. Louis, MO, USA). Mouse specific MLKL antibody (no. 28640) was from Cell Signaling Technology (Beverly, MA, USA). Recombinant mouse IL-4 (no. 574304), IL-10 (no. 575804) and M-CSF (no. 576404) were from Biolegend (San Diego, CA, USA). Recombinant mouse IFNα2 (no. 14-8312) and IFNγ (no. 34-8311 and no. 14-8311) were from Thermo Fisher Scientific (San Diego, CA, USA). Recombinant mouse TNFα (no. 410-MT) was from R & D Systems (Minneapolis, MN, USA). Polyinosinic-polycytidylic acid (poly(I:C)) (no. tlrl-pic), CL075 (no. tlrl-c75), and ODN 1585 (no. tlrl-1585) were from InvivoGen (San Diego, CA, USA). Z-VAD-FMK (no. A1902) was from ApexBio (Houston, TX, USA). Smac mimetic AZD5582 (no. CT-A5582) was from Chemietek (Indianapolis, IN, USA). Myristoylated autocamtide-2-related inhibitory peptide (Myr-AIP, no. BML-P212) was from Enzo (Farmingdale, NY, USA). Clarity Western ECL substrate (no. 1705060) was from Bio-Rad (Hercules, CA, USA). 

### 2.2. Cell Cultures

Primary alveolar epithelial cells (AECs, no. C57-6053), tracheal and bronchial epithelial cells (TBECs, no. C57-6033) and lung microvascular endothelial cells (no. C57-6011) from wild type C57BL/6 mice were obtained from Cell Biologics (Chicago, IL, USA). These primary lung cells were cultured in epithelial cell growth medium (M6621, Cell Biologics) or endothelial cell growth medium (M1168, Cell Biologics) and used for experiments within 3 passages. RAW264.7 murine macrophages (no. TIB-71) were from American Type Culture Collection (ATCC, Manassas, VA, USA) and cultured in DMEM medium supplemented with 10% fetal bovine serum (FBS).

### 2.3. Cell Viability, Lactate Dehydrogenase Release and Real Time Necroptosis Assays

Primary cells were seeded into 96-well plates, grown overnight to subconfluence and incubated with control PBS, IFNγ (1–50 ng/mL), IFNα2 (5–50 ng/mL), TNFα (25 ng/mL), LPS (100 ng/mL), poly(I:C) (0.5 μg/mL), CL075 (1 μg/mL), ODN1585 (1 μM), IL-4 (15 ng/mL), or IL-10 (15 ng/mL) for 24 h in the complete growth medium. The media were then aspirated, and the cells were washed once with PBC containing 1% FBS and treated in a 0–10 h period with mouse TNFα (15 ng/mL), poly(I:C) (0.2 µg/mL) or LPS (200 ng/mL) plus Smac mimetic AZD5582 (100 nM) and z-VAD-FMK (25 or 40 µM) to induce necroptosis in the phenol-free epithelial cell growth medium (no. PCS-300-040, ATCC, Manassas, VA, USA) (for lung epithelial cells) or in RPMI-1640 medium supplemented with 2.5% FBS (for endothelial cells). In some experiments, JAK kinase inhibitor-1 (1 µM), GSK872 (5 µM), GSK843 (2 μM), GW806742X (1–2 µM) or Myr-AIP (1 and 5 µM) were added to the medium. All the cells were incubated with the same amount of DMSO. Cell supernatants were saved, and the cell viability was assessed by MTS assay using CellTiter AQueous one solution reagent (no. G3580) according to the manufacturer’s instructions (Promega, Madison, WI, USA). Cell survival rates were calculated by comparison to individual control groups without treatment of necroptosis inducers and are presented as means ± SE from at least three independent experiments.

Lactate dehydrogenase (LDH) activities in cell supernatants were determined by using the LDH-Cytox™ assay kit (no. 426401, Biolegend, San Diego, CA, USA) according to the manufacturer’s protocol. Relative changes in the release of LDH are presented as fold ± SE of individual control groups without treatment of necroptosis inducers from at least three independent experiments.

Real time necroptosis is monitored by a 0–10 h period using the CellTox™ green cytotoxicity assay (no. G8742, Promega, Madison, WI, USA) according to the manufacturer’s protocol. The assay system uses an asymmetric cyanine dye that is excluded from viable cells but preferentially stains the dead cells’ DNA. The percentage of lytic cell death (green dye^+^ cells) was calculated by comparison to the maximal fluorescence induced by a lysis buffer in each group and is presented as mean ± SE from at least three independent experiments. 

### 2.4. DNA and High-Mobility Group Protein B1 Release Assays

After treatment with necroptosis inducers for 6 h, cell supernatants were saved, and the contents of released double-stranded DNA (dsDNA) were determined by a sensitive fluorescence dsDNA quantitation kit (no. 31007-T, Biotium, Fremont, CA, USA) according to the manufacturer’s protocol. The amounts of high-mobility group protein B1 (HMGB1) in cell supernatants were determined by using ELISA kits (no. MBS2701751, MyBioSource, San Diego, CA, USA). 

### 2.5. Real Time Measurement of ATP Release

Kinetic monitoring of ATP released from lytic cells was performed by using RealTime-Glo™ extracellular ATP assay kits (no. GA5010, Promega, Madison, WI, USA) according to the manufacturer’s protocol. The assay uses an optimized, bioluminescence detection chemistry to measure ATP that has been released into the cultured cell environment as a result of cell death and stress. The nonlytic, homogeneous ATP assay substrate was added directly to cells at the time of treatment with necroptosis inducers for a period of 0–10 h to assess continuous ATP release. 

### 2.6. Western Blot Analysis

Western blot analysis was performed as we described previously [31]. Briefly, cells were washed twice with ice-cold PBS and then lysed on ice in Nonidet P-40 lysis buffer (25 mM Tris-HCl, pH 7.5, 1% Nonidet P-40, 150 mM NaCl, 10 mM NaF, 1 mM Na3VO4, 1 mM phenylmethylsulfonyl fluoride, 10 mg/mL each leupeptin and aprotinin). Equal amounts of whole cell lysates were subjected to SDS-PAGE and then transferred to polyvinylidene difluoride membranes. The membranes were blocked 1 h at room temperature with 5% low fat milk in Tris-buffered saline plus 0.05% Tween 20, and then incubated 1 h at room temperature with primary antibodies against mouse MLKL (1:500, no. 28640, Cell Signaling Technology, Danvers, MA, USA) or vinculin (1:3000, no. V9131, Sigma, St. Louis, MO, USA), followed by 1 h incubation with anti-mouse (no. 7076) or anti-rabbit (no. 7074) HRP-conjugated secondary antibodies (Cell Signaling Technology, Beverly, MA, USA). Labelled proteins were detected with Bio-Rad Clarity Western ECL substrate (no. 1705061, Hercules, CA, USA). Relative changes in the expression levels of the proteins of interest were measured by densitometric analysis using ImageJ 1.47 software and normalized to vinculin and are represented as fold or percentage of control. 

### 2.7. RNA Isolation and Reverse Transcription (RT)-PCR 

Total RNA was isolated using the RNeasy RNA isolation kit (no. 74104) according to the manufacturer’s protocol (Qiagen, Valencia, CA, USA). Primers were designed by using Oligo6 software (Molecular Biology Insights) as follows: MLKL (expected product of 269 bp), 5′-CCG AAA GTG TTG GAA TAG TGA-3′ (forward) and 5Ȳ-GAG TGT TTC CGA ATG GTG TAG-3′ (reverse); and internal control glyceraldehyde-3-phosphate dehydrogenase (GAPDH) (608 bp), 5′-CGC TGA GTA CGT CGT GGA G-3′ (forward) and 5′-GAG GAG TGG GTG TCG CTG TTG-3′ (reverse). RT-PCR conditions were 50 °C for 30 min and 95 °C for 15 min, followed by 30 cycles of 94 °C for 30 s, 55 °C for 1 min, and 72 °C for 1 min, followed by 72 °C for 10 min. The relative changes in mRNA levels were measured by densitometric analysis and normalized to GAPDH.

### 2.8. SiRNAs and siRNA Transfection 

AllStars non-targeting negative control siRNA (no. 1027280) and mouse MLKL siRNA (no. SI00868812) were from Qiagen (Valencia, CA, USA). For siRNA transfection, cells were seeded into 96-well plates for 24 h to reach 50–70% confluence, and then siRNA was transfected into the cells in a final concentration of 20 nM by using Lipofectamine 2000 and Opti-MEM I reduced serum medium according to the manufacturer’s protocol (Invitrogen, Carlsbad, CA, USA). The silencing effects of siRNAs were confirmed by Western blot analysis.

### 2.9. Statistical Analysis

Data are expressed as mean ± SE. Statistical analyses were performed using GraphPad Prism 9 (GraphPad Software, La Jolla, CA, USA). Data were analyzed by using an unpaired, Student’s *t* test between means of two groups and one-way ANOVA for differences among means with three or more groups. Two-way ANOVA followed by Sidak multiple comparisons were performed to compare a response with 2 factors. Differences with *p* < 0.05 were considered statistically significant.

## 3. Results

### 3.1. IFNγ among Inflammatory Mediators Preferentially Promotes Necroptosis in Primary Alveolar Epithelial Cells

We determined whether lung epithelial cells that are pre-activated by pro-inflammatory mediators preferentially undergo necroptosis. As shown in Figure 1A, primary alveolar epithelial cells (AECs) from C57BL/6 mice were incubated with control PBS, IFNα, IFNγ, agonists of TLR3 (poly(I:C), TLR4 (LPS), TLR7 (CL075) and TLR9 (ODN1585), IL-4 or IL-10 for 24 h, and then treated for 6 h with control vehicle DMSO or three different necroptosis inducers including TNFα plus Smac mimetic and a pan-caspase inhibitor z-VAD-FMK (TSZ), poly(I:C) plus Smac mimetic and z-VAD-FMK (PSZ), or LPS plus Smac mimetic and z-VAD-FMK (LSZ) [32,33,34,35,36]. Cell viability was assessed by an MTS assay that determines the number of viable cells. We found that IFNγ remarkably enhanced cell death induced by TSZ, PSZ and LSZ (Figure 1A). Type-I IFNα also augmented cell death by TSZ and PSZ, but the effect was less than IFNγ. In contrast, TLR agonists, IL-4 or IL-10 demonstrated relatively minor effects (Figure 1A). 

The release of LDH is a hallmark of lytic cell death. Consistent with the cell survival data (Figure 1A), the release of LDH from primary AECs by necroptosis inducers TSZ, PSZ or LSZ was greatly enhanced by IFNγ pretreatment (Figure 1B). Type-I IFNα also enhanced the release of LDH induced by TSZ and PSZ, but the effect was much less than IFNγ. In contrast, TLR agonists, IL-4 or IL-10 had a minor effect. To verify the effect of IFNγ, AECs were pretreated with IFNγ in the presence of DMSO or a selective JAK kinase inhibitor [37] and then exposed to necroptosis inducers TSZ or PSZ for 6 h. We found that the enhanced release of LDH by TSZ or PSZ from IFNγ-treated AECs was essentially abolished by the selective JAK kinase inhibitor, indicating a requirement of IFNγ signaling (Figure 1C). The selective JAK kinase inhibitor did not significantly affect the release of LDH from control AECs in response to TSZ or PSZ. Noticeably, the releases of LDH from both control and IFNγ-treated AECs were blocked by a selective RIPK3 inhibitor GSK872 [36,38] (Figure 1C), indicating a key involvement of the RIPK3 kinase activity-dependent signaling pathway.

We next utilized the CellTox™ green cytotoxicity assay to monitor lytic cell death in real time. The assay system uses an asymmetric cyanine dye that is excluded from viable cells, so the viable cells produce no appreciable increases in fluorescence. When the dye binds DNA of compromised cells with impaired membrane integrity, its fluorescent properties are substantially enhanced. Therefore, the fluorescent signal produced by the dye binding to the dead-cell DNA is proportional to cytotoxicity and can be used to accurately monitor real-time lytic cell death. We found that IFNγ greatly accelerated mouse AEC necroptosis in real time by TSZ, PSZ and LSZ (Figure 2). The IFNγ-pretreated AECs had the most susceptibility to TSZ with ~80% of the cells being lysed 5h after TSZ treatment and all the cells lysed 6 h after TSZ treatment (Figure 2A). The IFNγ-promoted real-time necroptosis by TSZ was blocked by a selective JAK kinase inhibitor [37] or two structurally distinct specific RIPK3 inhibitors GSK872 or GSK843 [36,38], confirming the involvement of IFNγ-mediated and RIPK3 kinase activity-dependent signaling pathways (Figure 2B). Moreover, almost all the IFNγ-activated AECs underwent lysis after 10 h treatment with PSZ or LSZ (Figure 2C,D). Type-I IFNα also enhanced real-time necroptosis by TSZ and PSZ, but the effect was much less than IFNγ (Figure 2A,C). Conversely, TNFα, TLR-3, -4, -7, or -9 agonists had a minor effect on necroptosis in primary AECs (Figure 2). 

We determined the dose-dependent effects of type-II IFNγ and type-I IFNα on real-time necroptosis in primary AECs. We found that IFNγ at as low as 1 ng/mL markedly promoted real-time necroptosis induced by TSZ or PSZ (Figure 3A,C). IFNα generally had less of an effect. IFNα at 50 ng/mL enhanced TSZ-induced necroptosis (Figure 3A), while at 10 and 50 ng/mL, it significantly promoted PSZ-induced necroptosis (Figure 3C). The dose-dependent effects of IFNγ on necroptosis were verified by measuring the release of LDH from AECs (Figure 3B,D). Moreover, the LSZ-induced real-time necroptosis was dose-dependently accelerated by IFNγ, while IFNα essentially did not affect the necroptosis (Figure 3E). Taken together, our findings indicate that IFNγ preferentially promotes necroptosis in primary AECs. 

### 3.2. IFNγ Preferentially Promotes Necroptosis in Primary Airway Epithelial Cells

We next assessed if IFNγ also promotes necroptosis in primary airway epithelial cells. Tracheal and bronchial epithelial cells (TBECs) from wild type C57BL/6 mice were utilized for this study. We found that IFNγ greatly accelerated TBEC necroptosis induced by TSZ, PSZ and LSZ (Figure 4). IFNγ-activated TBECs had the most susceptibility to TSZ with all cells lysed 5 h after TSZ treatment (Figure 4A). Almost all the IFNγ-activated TBECs underwent lysis after 9 h of treatment with PSZ or LSZ (Figure 4C,E). Dose-dependent real-time necroptosis studies indicate that IFNγ at 1 ng/mL is sufficient to achieve 100% cell lysis after 8 h treatment with TSZ or PSZ (Figure 4B,D). Although type-I IFNα at 50 ng/mL could enhance necroptosis by TSZ and PSZ, the effect was less than that of IFNγ at 1 ng/mL (Figure 4B,D). In contrast, TNFα, TLR-3, -4, -7, or -9 agonists generally had a relatively minor effect on the necroptosis in primary airway epithelial cells. Collectively, these findings indicate that IFNγ also preferentially promotes necroptosis in primary airway epithelial cells.

In contrast to lung epithelial cells, we found that IFNα, IFNγ, TNFα, as well as TLR agonists did not significantly promote necroptosis in response to TSZ, PSZ or LSZ in mouse primary lung microvascular endothelial cells (Appendix A).

### 3.3. IFNγ Greatly Accelerates the Release of DAMPs from AECs

Necroptosis causes cell membrane rupture, which triggers and amplifies inflammation through the release of DAMPs, such as ATP, the high-mobility group protein B1 (HMGB1), nucleic acids, IL-1 family cytokines and S100 proteins [11,12]. We monitored kinetic ATP release from lytic cells by using RealTime-Glo™ extracellular ATP assay. We found that IFNγ greatly accelerated the release of ATP from AECs in response to TSZ, with a peak at 3 h after TSZ treatment (Figure 5A). The basal level of HMGB1 was detectable in cell supernatants from resting AECs. At 6 h of TSZ treatment, the levels of HMGB1 were not significantly increased in control AECs but were markedly increased in IFNγ-activated AECs (Figure 5B). We further found that IFNγ also accelerated the release of dsDNA from AECs in response to TSZ (Figure 5C). The level of dsDNA was not detectable in cell supernatants from resting AECs and control AECs treated with TSZ for 6 h (Figure 5C). Collectively, these findings demonstrate that IFNγ greatly accelerates the release of DAMPs from primary AECs.

### 3.4. MLKL Is Predominantly Induced by IFNγ in Primary Lung Epithelial Cells and MLKL Silencing or Inhibition Largely Suppresses the IFNγ-Promoted Necroptosis 

MLKL is a central effector downstream of RIPK3 kinase [13,14]. We found that MLKL mRNAs were readily induced by IFNγ in primary AECs, TBECs and Raw264.7 macrophages (Figure 6A). We further found that MLKL protein levels were predominantly upregulated by IFNγ among the tested inflammatory mediators, although type-I IFNα had an effect (Figure 6B–D). In contrast, TNFα, IL-4, TLR-3, -4, -7 or -9 agonists had a minor effect. Interestingly, MLKL expression could be synergistically induced by IFNγ plus TNFα (Figure 6C). 

We knocked down MLKL and determined the effect on the IFNγ-promoted necroptosis in primary AECs. Real-time necroptosis analysis revealed that the enhanced necroptosis in IFNγ-activated AECs was significantly and markedly inhibited by MLKL silencing, although the inhibitory effect was reduced at a later time when necroptosis occurred (after 6 h of TSZ treatment) (Figure 6E,F). We next determined the effect of a selective murine MLKL inhibitor GW806742X on the IFNγ-promoted necroptosis. GW806742X inhibits necroptosis by preventing MLKL membrane translocation [39]. We found that the enhanced necroptosis in IFNγ-activated AECs was inhibited 98%, 92%, 90%, 87%, or 85% from 2 to 6 h during TSZ treatments by GW806742X (1 or 1.5 µM), respectively (Figure 6G). However, GW806742X became less effective, especially in the later stages of necroptosis (after 7 h of TSZ treatment). GW806742X at concentrations of 1 and 1.5 μM appeared to have slightly better inhibitory effects than that of 2 µM (Figure 6G). Interestingly, necroptosis in IFNγ-activated AECs was no longer inhibited by GW806742X at three different doses (1, 1.5 or 2 µM) after 24 h of TSZ treatment, whereas necroptosis in control AECs was sustainably and markedly suppressed by GW806742X at the time point (Figure 6H). 

Ca^2+^-calmodulin-dependent protein kinase-II (CaMKII) has been identified as a new RIPK3 effector mediating myocardial necroptosis [40]. We treated IFNγ-activated AECs with a specific CaMKII myristoylated peptide inhibitor (Myr-AIP) at two different concentrations (1 and 5 µM) that have been shown to effectively inhibit CaMKII activities in vitro and in cultured cells [41,42] and assessed the effects on TSZ-induced necroptosis in primary AECs. We found that inhibition of CaMKII activity with Myr-AIP led to a small but statistically significant reduction in the IFNγ-promoted necroptosis at 3 h (30% inhibition), 4 h (25% inhibition) or 5 h (14% inhibition) of TSZ treatment (Figure 6G). Myr-AIP did not significantly affect necroptosis after 6 h of TSZ treatment. Interestingly, Myr-AIP plus GW806742X did not lead to an additive inhibitory effect on necroptosis compared with GW806742X alone (Figure 6G), suggesting that MLKL and CaMKII may signal and function through the same necroptosis pathway in AECs.

## 4. Discussion

The present study provides compelling evidence that IFNγ preferentially among various inflammatory mediators promotes necroptosis in primary alveolar and airway epithelial cells but not in lung microvascular endothelial cells. This process involves the IFNγ-mediated upregulation of necroptosis effector MLKL. Unlike apoptosis, necroptosis causes cell membrane rupture, which triggers and amplifies inflammation through the release of DAMPs, such as ATP, HMGB1, nucleic acids, IL-1 family cytokines and S100 proteins [11,12]. Our findings also demonstrate that IFNγ greatly accelerates the release of ATP, HMGB1 and dsDNA from primary lung epithelial cells undergoing necroptosis. DAMPs are detected by pattern recognition receptors that activate immune responses by inducing the expression of cytokines and chemokines. HMGB1 is a highly conserved ubiquitous protein present in the nucleus and cytoplasm of nearly all cell types and is the prototypic DAMP molecule [43]. The inflammatory functions of HMGB1 are mediated through the receptor for advanced glycation end products (RAGE), TLR2, TLR4, and TLR9 [44,45]. ATP activates P2 purinergic receptors, through which it is involved in multiple pulmonary disorders [46,47], while dsDNA including mitochondrial DNA triggers an innate immune response through DNA sensors [48,49]. On the basis of these results, we posit that enhanced necroptosis in IFNγ-activated lung epithelial cells not only promotes cell death but also accelerates the release of immune-stimulatory DAMPs, through which they may contribute to the exaggerated tissue injury, inflammation and lung dysfunction.

IFNγ can be produced by innate (NK, NKT, antigen-presenting cells, neutrophils) and adaptive (CD4^+^, CD8^+^, B) immune cells [50,51] and is often elevated in patients with autoimmune diseases [50,52,53] and infections with respiratory viruses such as SARS-CoV-2 [54,55,56,57,58]. A prospective cohort study identified IFNγ, angiopoietin1/2, IL-6, and plasminogen activator inhibitor-1 as four critical biomarkers for the “reactive” phenotype group of ARDS, which has higher mortality than a less “inflamed” phenotypic group [59]. Prior studies demonstrate that neutralization of IFNγ or deficiency of IFNγ receptor confer protection against ALI and/or lethality induced by LPS [60,61,62], sepsis/systemic inflammatory response syndrome [62,63,64,65], hyperoxia/mechanical ventilation [66,67], and respiratory viral infection [68,69,70,71]. Neutralization of both IFNγ and TNFα additionally protects against mortality by SARS-CoV2 infection in mice [72]. Administration of recombinant IFNγ or local IFNγ induction greatly accelerates ALI following LPS challenge in mice [61]. It has been demonstrated that necrosis (including necroptosis) rather than apoptosis is the dominant form of AEC death in LPS-induced ALI in mice [73]. However, IFNγ alone is unable to induce ALI in mice [61]. Collectively, these data indicate that IFNγ appears to play an important role in ALI, but the underlying molecular mechanisms by which this mediator regulates lung injury remain unclear. The present study has now identified a mechanistic link between IFNγ and the enhanced necroptosis of activated lung epithelial cells and expanded understanding of the contribution of IFNγ to the pathogenesis of ALI. High levels of pulmonary IFNγ may cause a rapid initiation and fast progression of ALI that is often observed in ARDS [2,74] where necroptosis signals are triggered by mechanical ventilation, endotoxin inhalation, infections and trauma [7,8,10].

It has been shown that extended incubation (24 to 72 h) of IFNγ induces necroptosis in murine embryo fibroblasts (MEFs) deficient of NF-κB signaling (*rela*^-/-^) or Fas-associated death domain (*fadd*^-/-^) or when caspases are inhibited [75,76]. NF-κB, FADD and caspases are negative intracellular regulators of necroptosis [13,14], thus IFNγ facilitates necroptosis in the sensitized MEFs. However, IFNγ had a minor effect on the cell survival of primary normal MEFs [75,76]. Conversely, Lee et al. reported that IFNγ slightly promoted a reduction in MLKL and inhibited the phosphorylation of MLKL in mouse splenocytes treated with zVAD and TNFα, in which the mechanism was unknown [77]. Consistent with the effect on primary normal MEFs [75,76], we found that 24 h incubation of IFNγ alone had no effect on cell survival but slightly reduced LDH release from primary normal mouse AECs (Appendix A). We further found that IFNγ markedly promoted the susceptibility of lung epithelial cells to various necroptosis inducers. Hence, we posit that IFNγ has a minor effect on cell survival in primary normal cells but, in a cell type-dependent manner, can greatly promote necrotic cell death in necroptosis-sensitized cells and cells receiving necroptosis signals. The IFNγ-promoted necroptosis of lung epithelial cells was blocked by a selective JAK kinase inhibitor [37] or two structurally distinct specific RIPK3 kinase inhibitors GSK872 or GSK843 [36,38]. These results indicate a requirement of IFNγ signaling and support a key role of RIPK3 kinase activity in the initiation of necroptosis [35]. We further found that the mRNA and protein levels of necroptosis effector MLKL were predominantly upregulated by IFNγ in lung epithelial cells and that the enhanced necroptosis in the cells was significantly and markedly inhibited by MLKL silencing. Moreover, the enhanced necroptosis in IFNγ-activated AECs was almost completely abolished by preventing MLKL membrane translocation with a selective murine MLKL inhibitor GW806742X [39] during the early stages of necroptosis (from 2 to 6 h of TSZ treatments). These findings indicate that the induction of MLKL is a critical determinant of enhanced necrotic cell death that occurs in IFNγ-activated lung epithelial cells. We recently showed that IFNγ, LPS, or poly(I:C) induced MLKL expression and promoted necroptosis in mouse bone marrow-derived macrophages [78]. However, MLKL was barely induced by LPS and poly(I:C) in primary lung epithelial cells and neither LPS nor poly(I:C) promoted lung epithelial cell necroptosis. Similarly, we found that IFNγ did not promote necroptosis in lung microvascular endothelial cells, which may be attributed to the inability of IFNγ to induce MLKL expression. Cell type-dependent regulation of MLKL expression by IFNγ and the underlying mechanism merit further investigation. Interestingly, we found that GW806742X became less effective in suppressing necroptosis during the later stages of necroptosis (after 7 h of TSZ treatment). GW806742X inhibits necroptosis by retarding MLKL translocation to membranes [39]. Our observation could be interpreted as that membrane translocation of MLKL in IFNγ-activated lung epithelial cells may not be effectively blocked by GW806742X, possibly due to a modification of MLKL. Other mechanisms beyond the participation of MLKL could likewise contribute to the enhanced necroptosis at the later stages in IFNγ-activated lung epithelial cells. 

CaMKII is another necroptosis effector downstream of RIPK3, mediating myocardial necroptosis [40]. We found that inhibition of CaMKII activity with a specific myristoylated peptide inhibitor (Myr-AIP) [41,42] led to a small but statistically significant reduction in the IFNγ-promoted necroptosis at 3 h (30% inhibition) and 4 h (25% inhibition) of TSZ treatment, where the enhanced necroptosis was suppressed more than 90% by the MLKL inhibitor GW806742X [39]. Furthermore, Myr-AIP plus GW806742X did not lead to an additive inhibitory effect on necroptosis compared with GW806742X alone. A recent report shows that CaMKII regulates necroptosis downstream of MLKL in smooth muscle cells [41]. Hence, our findings indicate that CaMKII contributes to a small proportion of necroptosis early after induction of the process in IFNγ-activated lung epithelial cells and may likely mediate necroptosis downstream of MLKL as well.

In conclusion, we provide compelling evidence showing that IFNγ preferentially among various inflammatory mediators promotes necroptosis and accelerates the release of DAMPs in primary alveolar and airway epithelial cells but not in lung microvascular endothelial cells. We further show that MLKL is predominantly induced by IFNγ, contributing to the enhanced necroptosis in lung epithelial cells. Our findings provide a mechanistic link and a novel interpretation of the contribution of IFNγ to the pathogenesis of cellular perturbations that may contribute to ALI.

## Figures and Tables

**Figure 1 cells-11-00563-f001:**
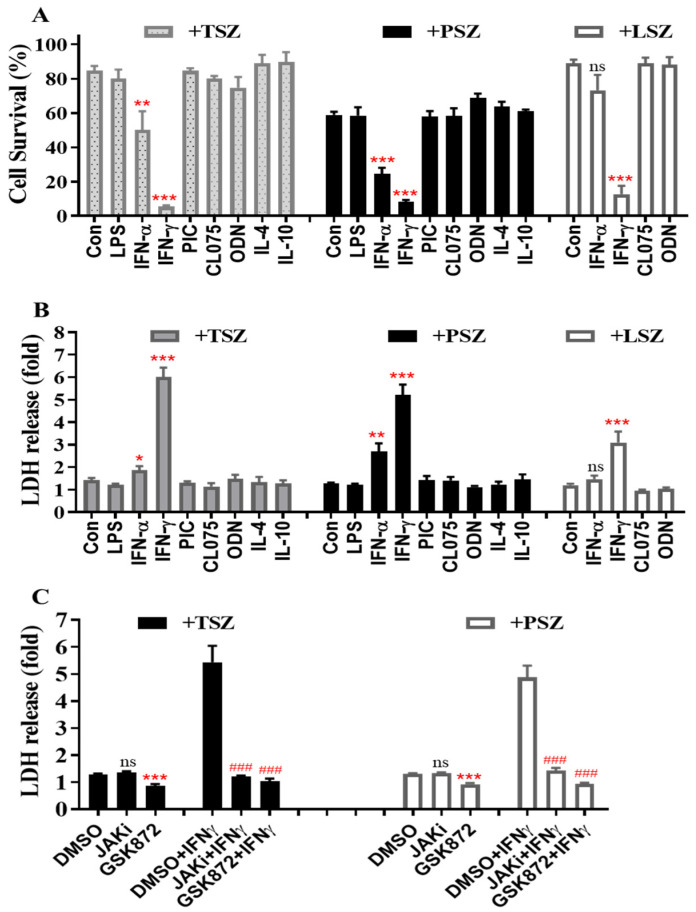
IFNγ preferentially promotes cell death and the release of LDH induced by various necroptosis inducers in primary AECs. Mouse primary AECs were incubated with LPS (100 ng/mL), IFNα2 (50 ng/mL), IFNγ (50 ng/mL), poly(I:C) (PIC, 0.5 μg/mL), CL075 (1 μg/mL), ODN1585 (1 μM), IL-4 (15 ng/mL), or IL-10 (15 ng/mL) for 24 h, then treated for 6 h with TNFα (15 ng/mL) plus 100 nM Smac mimetic AZD5582 and 25 µM z-VAD-FMK (TSZ), poly(I:C) (0.2 µg/mL) plus 100 nM Smac mimetic AZD5582 and 40 µM z-VAD-FMK (PSZ), or LPS (200 ng/mL) plus 100 nM Smac mimetic AZD5582 and 25 µM z-VAD-FMK (LSZ). (**A**) AEC viability was assessed by MTS assay using CellTiter AQueous reagent and cell survival rates were calculated by comparison to individual control groups without treatment of necroptosis inducers and are presented as means ± SE. Con.+TSZ or PSZ, n = 12; IFNγ+TSZ or PSZ, n = 10; all other treatments, n = 4–8. ** *p* < 0.01; *** *p* < 0.001 versus control (Con.) in each treatment group. NS, no significance. (**B**) LDH activities in cell supernatant were determined. Relative changes in LDH are presented as fold ± SE of individual control groups without treatment of necroptosis inducers. Con. or IFNγ+TSZ, n = 10; Con. or IFNγ+PSZ, n = 12; all other treatments, n = 4–7. * *p* < 0.05; ** *p* < 0.01; *** *p* < 0.001 versus control (Con.) in each group. NS, no significance. (**C**) Mouse AECs were incubated without or with IFNγ (50 ng/mL) in the presence of DMSO or JAK kinase inhibitor-1 (JAKi, 1 μM) for 24 h, washed and then treated for 6 h with necroptosis inducers TSZ or PSZ in the presence of DMSO, GSK872 (5 μM) or JAKi (1 μM). LDH activities in cell supernatant were determined and the relative changes are presented as fold ± SE of individual control groups without treatment of necroptosis inducers. TSZ group: DMSO or DMSO+IFNγ, n = 7; others, n = 3–4. PSZ group: DMSO or DMSO+IFNγ, n = 9; others, n = 5. *** *p* < 0.001 versus DMSO in each group. ### *p* < 0.001 versus DMSO+IFNγ in each group. NS, no significance. One-way ANOVA was performed in (**A**–**C**).

**Figure 2 cells-11-00563-f002:**
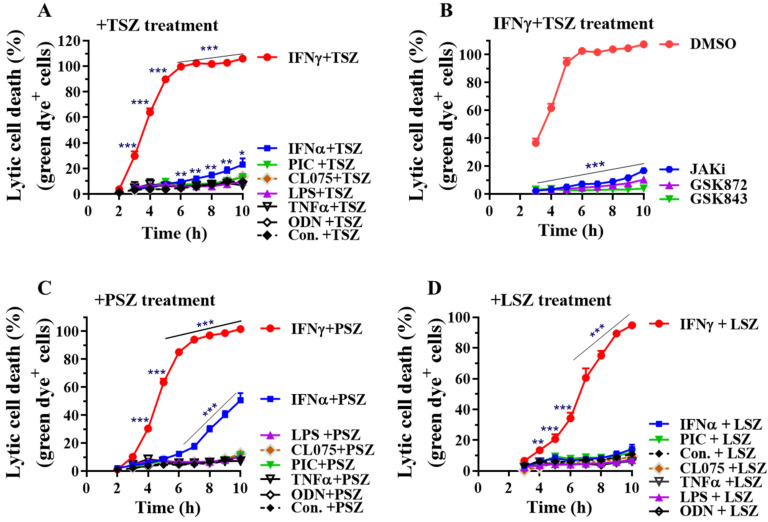
IFNγ preferentially promotes AEC necroptosis by real-time lytic cell death analysis. Mouse primary AECs were incubated with control PBS (Con.), IFNα2 (50 ng/mL), IFNγ (50 ng/mL), TNFα (25 ng/mL), LPS (100 ng/mL), poly(I:C) (PIC, 0.5 μg/mL), CL075 (1 μg/mL) or ODN1585 (1 μM) for 24 h, then treated in a 0–10 h period with DMSO, or TNFα (15 ng/mL) plus 100 nM Smac mimetic AZD5582 and 25 µM z-VAD-FMK (**A**, +TSZ), poly(I:C) (0.2 µg/mL) plus 100 nM Smac mimetic and 40 µM z-VAD-FMK (**C**, +PSZ) or LPS (200 ng/mL) plus 100 nM Smac mimetic AZD5582 and 25 µM z-VAD-FMK (**D**, +LSZ) in the presence of an asymmetric cyanine dye. (**B**) Mouse AECs were incubated without or with IFNγ (50 ng/mL) in the presence of DMSO or JAK kinase inhibitor-1 (JAKi, 1 μM) for 24 h, washed and then subjected to TSZ-induced real-time necroptosis assay as described above in the presence of DMSO, GSK872 (5 μM), GSK843 (2 μM) or JAKi (1 μM). The percentage of lytic cell death (green dye^+^ cells) was calculated by comparison to the maximal fluorescence induced by a lysis buffer in each group and presented as mean ± SE. In panel (**A**): Con.+TSZ, n = 16; IFNγ+TSZ, n = 15; IFNα+TSZ, n = 11; other treatments, n = 4–6. In panel (**B**): DMSO, n = 10; others, n = 5–7. In panel (**C**): Con.+PSZ, n = 15; IFNγ+PSZ, n = 14; IFNα+PSZ, n = 10; other treatments, n = 4–5. In panel (**D**): Con.+LSZ, n = 8; IFNγ or IFNα+LSZ, n = 6; other treatments, n = 4–5. Two-way ANOVA and unpaired Student’s *t* test were performed in (**A**–**D**). * *p* < 0.05; ** *p* < 0.01; *** *p* < 0.001 versus control (Con.) or DMSO in each group. The orders of each agonist-induced percentage of lytic cell death were listed on the right in panels (**A**,**C**,**D**).

**Figure 3 cells-11-00563-f003:**
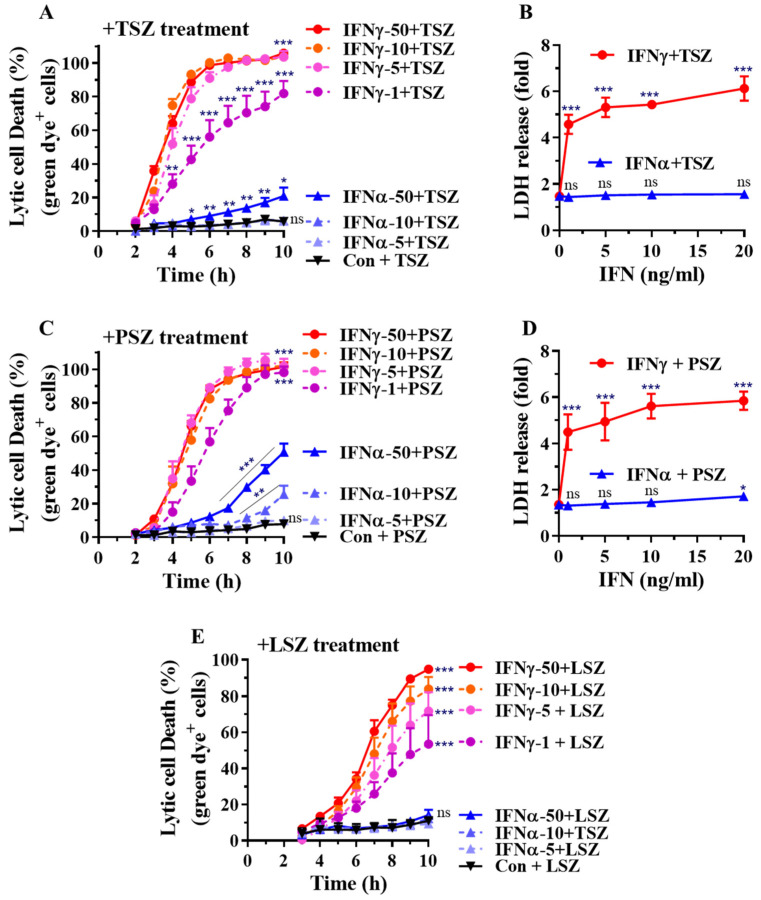
Dose-dependent effects of IFNγ and IFNα on AEC necroptosis. (**A**,**C**,**E**) Mouse primary AECs were incubated with control PBS (Con.), IFNα2 at 5, 10, or 50 ng/mL, or IFNγ at 1, 5, 10, or 50 ng/mL for 24 h. The cells were then subjected to real-time necroptosis assay as described in Figure 2 in response to TSZ (**A**), PSZ (**C**) or LSZ (**E**). The percentage of lytic cell death (green dye^+^ cells) was calculated by comparison to the maximal fluorescence induced by a lysis buffer in each group and is presented as mean ± SE. In panel (**A**): Con., IFNγ-1, IFNγ-10, IFNγ-50, or IFNα-50+TSZ, n = 10; other treatments, n = 4–7. In panel (**C**): Con., IFNγ-50, or IFNα-50+PSZ, n = 10; other treatments, n = 4–7. In panel (**E**): Con.+LSZ, n = 8; IFNα-5+LSZ, n = 3; all other treatments, n = 6. Two-way ANOVA and unpaired Student’s t test were performed in (**A**,**C**,**E**). * *p* < 0.05; ** *p* < 0.01; *** *p* < 0.001 versus control (Con.) in each treatment group. NS, no significance. The orders of each agonist-induced percentage of lytic cell death were listed on the right in each panel. (**B**,**D**) Mouse primary AECs were incubated with IFNα2 or IFNγ at 0, 1, 5, 10, or 20 ng/mL for 24 h, then treated for 6 h with necroptosis inducers TSZ (**B**) or PSZ (**D**). LDH activities in cell supernatant were determined and the relative changes are presented as fold ± SE of individual control groups without treatment of necroptosis inducers. Panel (**B**), n = 4–7; panel (**D**), n = 3–6. One-way ANOVA was performed in (**B**,**D**). *** *p* < 0.001 versus control (0 IFN) in each treatment group. NS, no significance.

**Figure 4 cells-11-00563-f004:**
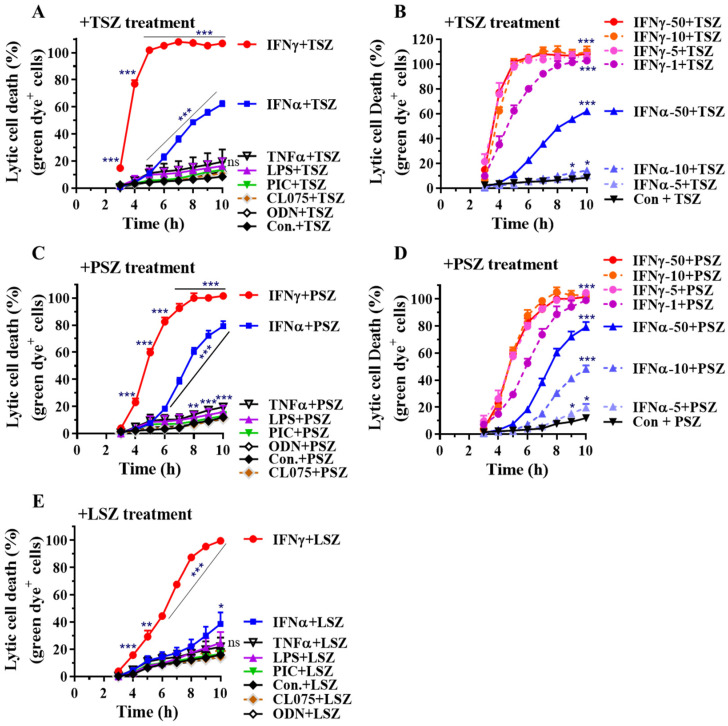
IFNγ predominantly promotes TBEC necroptosis by real time lytic cell death analysis. (**A**,**C**,**E**) Mouse primary TBECs were incubated with PBS (Con.), IFNα2 (50 ng/mL), IFNγ (50 ng/mL), TNFα (25 ng/mL), LPS (100 ng/mL), poly(I:C) (PIC, 0.5 μg/mL), CL075 (1 μg/mL) or ODN1585 (1 μM) for 24 h. (**B**,**D**) Mouse primary AECs were incubated with control PBS (Con.), IFNα2 at 5, 10, or 50 ng/mL, or IFNγ at 1, 5, 10, or 50 ng/mL for 24 h. The cells were then treated for 0 to 10 h with necroptosis inducers TSZ (**A**,**B**), PSZ (**C**,**D**) or LSZ (**E**) in the presence of an asymmetric cyanine dye. The percentage of lytic cell death (green dye^+^ cells) was calculated by comparison to the maximal fluorescence induced by a lysis buffer in each group and is presented as mean ± SE. In panel (**A**): Con. or IFNγ+TSZ, n = 7–8; all other treatments, n = 3. In panel (**B**): Con. or IFNγ-50+TSZ, n = 7–8; IFNγ-1 or IFNγ-5+TSZ, n = 5; other treatments, n = 3–4. In panel (**C**): Con. or IFNγ+PSZ, n = 7–8; all other treatments, n = 3. In panel (**D**): Con. or IFNγ-50+TSZ, n = 7–8; all other treatments, n = 3–4. In panel (**E**): n = 4–5. Two-way ANOVA was performed in (**A–E**). * *p* < 0.05; ** *p* < 0.01; *** *p* < 0.001 versus control (Con.) in each treatment group. NS, no significance. The orders of each agonist-induced percentage of lytic cell death were listed on the right in each panel.

**Figure 5 cells-11-00563-f005:**
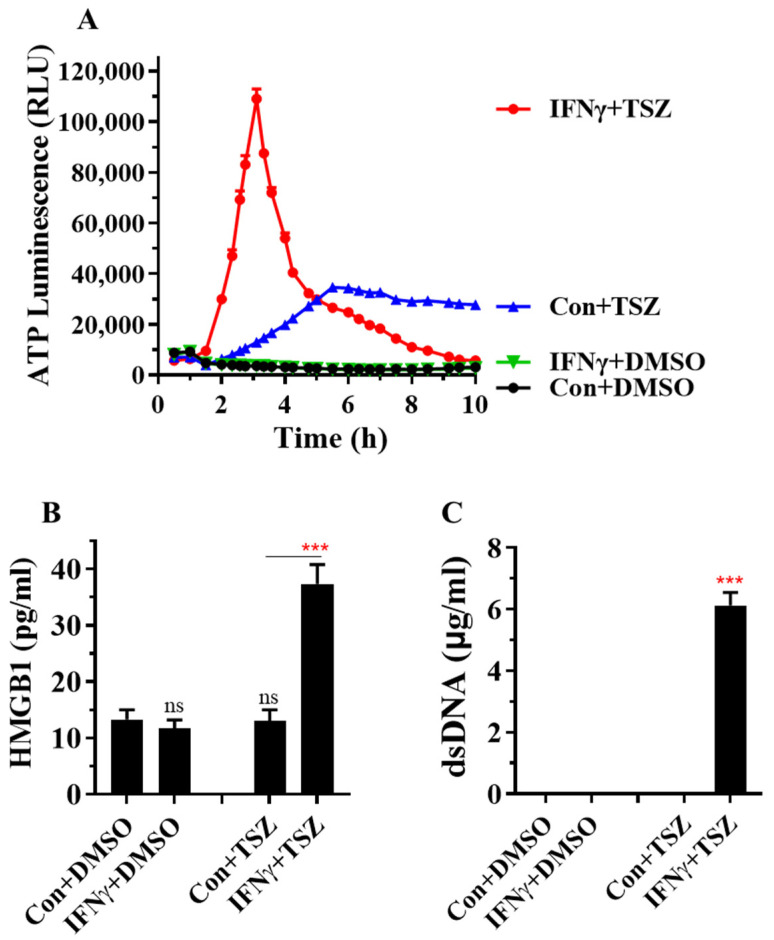
IFNγ greatly accelerates the release of DAMPs from AECs. (**A**) Mouse primary AECs were incubated with PBS (Con.) or IFNγ (50 ng/mL) for 24 h, then treated for 0 to 10 h with DMSO or necroptosis inducer TSZ. The release of ATP from AECs was continuously monitored by using RealTime-Glo™ extracellular ATP assay kits, and the levels of ATP luminescence are presented as means ± SE (n = 8). (**B**,**C**) Mouse primary AECs were incubated with PBS (Con.) or IFNγ (50 ng/mL) for 24 h, then treated for 6 h with DMSO or necroptosis inducer TSZ. The levels of HMGB1 and dsDNA in cell supernatant were determined and are presented as means ± SE (n = 4). Student’s *t* test was performed in (**B**,**C**). *** *p* < 0.001 versus control (Con.) in each treatment group. NS, no significance.

**Figure 6 cells-11-00563-f006:**
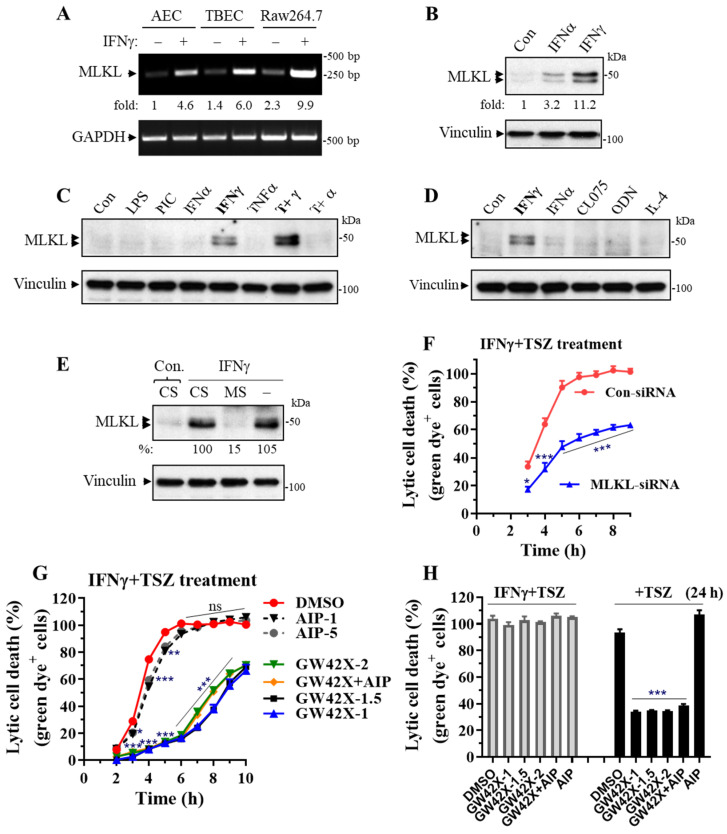
MLKL is predominantly induced by IFNγ in primary lung epithelial cells, and MLKL silencing or inhibition largely suppresses the IFNγ-promoted necroptosis. (**A**) RT-PCR showing the induction of MLKL by IFNγ. Mouse primary AECs and TBECs or Raw264.7 macrophages were treated with IFNγ (50 ng/mL) for 24 h, and mRNA levels of MLKL and GAPDH are shown in a representative RT-PCR. (**B**–**D**) Western blots showing the expression of MLKL. Primary mouse AECs were incubated with control PBS (Con.), IFNα2 (50 ng/mL), IFNγ (50 ng/mL), LPS (100 ng/mL), poly(I:C) (PIC, 0.5 μg/mL), TNFα (25 ng/mL), TNFα (25 ng/mL) plus IFNγ (50 ng/mL) (T+γ), TNFα (25 ng/mL) plus IFNα (50 ng/mL) (T+α), CL075 (1 μg/mL), ODN1585 (1 μM) or IL-4 (15 ng/mL) for 24 h. Equal amounts of cell lysates were subjected to Western blot analysis. Results represent the findings of 3 independent experiments. (**E**) MLKL knockdown by siRNA. Primary mouse AECs were transfected with 20 nM control siRNA (CS) or MLKL siRNA (MS) for 6 h, then treated with control PBS (Con) or IFNγ (25 ng/mL) for 24 h. Equal amounts of cell lysates were subjected to Western blots. Results represent the findings of 3 independent experiments. (**F**) MLKL silencing largely suppresses the IFNγ-promoted necroptosis. Primary mouse AECs were transfected with control siRNA or MLKL siRNA for 6 h, incubated with IFNγ (25 ng/mL) for 24 h, then subjected to TSZ-induced real-time necroptosis assay as described in Figure 2. * *p* < 0.05; ** *p* < 0.01; *** *p* < 0.001 versus control siRNA in each time point (n = 4). (**G**,**H**) Effects of MLKL or CaMKII inhibitors on necroptosis. Mouse AECs were incubated without or with IFNγ (25 ng/mL) for 24 h, then subjected to TSZ-induced real-time necroptosis assay as described above in the presence of DMSO, GW806742X at 1 μM (GW42X-1), 1.5 μM (GW42X-1.5), 2 μM (GW42X-2) or 2 μM plus 1 μM Myr-AIP (GW42X+AIP), or Myr-AIP at 1 μM (AIP-1) or 5 μM (AIP-5) over a 0–10 h period (**G**) or at 24 h of TSZ treatment (**H**). In panel (**G**): DMSO, n = 12; GW42X-1, n = 9; GW42X-1.5, n = 6; all others, n = 4. In panel (**H**): DMSO, n = 8; GW42X-1 or GW42X-1.5, n = 6; all others, n = 4. * *p* < 0.05; ** *p* < 0.01; *** *p* < 0.001 versus DMSO in each group. NS, no significance. Two-way ANOVA was performed in (**F**,**G**) and one-way ANOVA was performed in (**H**).

## Data Availability

All datasets generated for this study are included in the manuscript and Appendix A.

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
