# Peer review of "Interferon-γ Preferentially Promotes Necroptosis of Lung Epithelial Cells by Upregulating MLKL"

_cells, 2022, doi:10.3390/cells11030563_

Round 1

Reviewer 1 Report

Qin Hao et al. presented that the IFNg priming potentiates the necroptosis induced by various stimuli in alveolar and airway epithelial cells and further showed IFNg upregulates the expression of MLKL, a key regulator for necroptosis.

The study is interesting and might be contribute to interpret the impact of IFNg to the pathogenesis of necroptosis-related disease, as the authors suggested.

I have following comments

  1. The impact of IFNg on necroptosis seems differ depend on cell type or cellular context. Roshan et al. (2011, 2013) suggested IFNg itself has an ability to induce necroptosis and SH Lee et al. (2017) showed IFNg regulate necroptosis through the downregulation of MLKL, which is somewhat contrary to the contents of this manuscript. Moreover, the authors also showed the differential effect of IFNg among the cells studied here.

Therefore, the authors should mention their views on the role of IFNg in a different context in the discussion section.

  1. The authors showed IFNg treatment does not facilitate necroptosis induction in microvascular endothelial cells (Figure S1). I think this is also important to explain why IFNg have no effect on certain cells. Since the authors claim that IFNg priming potentiates the necroptosis by upregulation of MLKL expression in alveolar and airway epithelial cells, the result of the MLKL expression on IFNg-treated microvascular endothelial cells will strengthen the author’s message.
  2. In Figure 6G, the cell death still occurs in the presence of GW42x. What type of cell death occurs here? Just delayed necroptosis or IFNg mediated-other type of cell death? This issue should be added in the discussion section for readers.  

References

Roshan (2013) Proc Natl Acad Sci U S A. 2013 Aug 13;110(33):E3109-18.

doi: 10.1073/pnas.1301218110.

Roshan (2011), Mol Cell Biol. 2011 Jul;31(14):2934-46. doi: 10.1128/MCB.05445-11. 

SH Lee (2017). Sci Rep. 2017 Aug 31;7(1):10133. doi: 10.1038/s41598-017-09767-0.

Reviewer 2 Report

The authors investigated about the contribution of interferon-ϒ into the pathogenesis of ALI. The rational behind the study was clear and straight forward. The manuscript is almost well written. Overall the topic could be interesting but many details are not clear.

I recommend that the paper be accepted with minor revision:

a)  The authors should better emphasize the conclusion in abstract section.

b) In the introduction section, little previous evidence is provided about the importance of ALI in daily life. Incorporating comparisons with other studies would increase the strength of the paper. Please refer to doi: 10.3390/ijms22115533; 10.1586/177666X.5.1.63.

c) There are some minor grammar issues that should be fixed in order to aid the accessibility of the results to the reader.

Round 2

Reviewer 1 Report

Although the authors did not present new data about the MLKL expression on IFNg-treated microvascular endothelial cells, addition of new comments in discussion section made the manuscript better shape than the previous version. I have no other comment.